# Retrieval of Harmonized LAI Product of Agricultural Crops from Landsat OLI and Sentinel-2 MSI Time Series

Jiří Tomíček [1,2,*], Jan Mišurec [1], Petr Lukeš [3] and Markéta Potůčková [2]

1   Gisat Ltd., Milady Horákové 57, 170 00 Prague, Czech Republic
2   Department of Applied Geoinformatics and Cartography, Faculty of Science, Charles University, Albertov 6, 128 43 Prague, Czech Republic
3   Global Change Research Institute, Czech Academy of Sciences, Bělidla 986/4a, 603 00 Brno, Czech Republic
*   Correspondence: jiri.tomicek@gisat.cz

**Abstract:** In this study, an approach for the harmonized calculation of the Leaf Area Indices (LAIs) for agronomic crops from Sentinel-2 MSI and Landsat OLI multispectral satellite data is proposed in order to obtain a dense seasonal trajectory. It was developed and tested on dominant crops grown in the Czech Republic, including winter wheat, spring barley, winter rapeseed, alfalfa, sugar beet, and corn. The two-step procedure harmonizing Sentinel-2 MSI and Landsat OLI spectral data began with deriving NDVI, MSAVI, and NDWI_1610 vegetation indices (VIs) as proxy indicators of green biomass and foliage water content, the parameters contributing most to a stand's spectral response. Second, a simple linear transformation was applied to the resulting VI values. The regression model itself was built on an artificial neural network, then trained on PROSAIL simulations data. The LAI estimates were validated using an extensive dataset of in situ measurements collected during 2017 and 2018 in the lowlands of the Central Bohemia Region. Very strong agreement was observed between LAI estimates from both Sentinel-2 MSI and Landsat OLI data and independent ground-based measurements (*r* between 0.7 and 0.98). Very good results were also achieved in the mutual comparison of Sentinel-2 and Landsat-based LAI datasets (*rRMSE* < 20%, *r* between 0.75 and 0.99). Using data from all currently available Sentinel-2 (A/B) and Landsat (8/9) satellites, a dense harmonized LAI time series can be created with high potential for use in precision agriculture.

**Keywords:** Sentinel-2; Landsat; leaf area index; harmonization; vegetation index; PROSAIL; radiative transfer; artificial neural network; time series



## 1. Introduction

A major and undeniable advantage of satellite remote sensing over imaging by aircraft or UAV systems is its capacity for acquiring a consistent time series of observations [1]. That, in turn, provides an additional type of information for interpretation, which is the evolution of surface reflectance over time [2]. This is crucial and potentially valuable information for land surfaces behaving dynamically over time, such as agricultural crops or forests [3]. For some applications (e.g., crop yield estimation), the temporal resolution of a remote sensing system may even be more important than the spatial or spectral resolution [3–6].

Among the most interesting of the dynamically changing vegetation parameters for the purposes of, for example, precision agriculture, is the so-called Leaf Area Index (LAI) [7,8]. It describes the amount of leaf biomass in a stand, which directly influences the resulting yield [7,8]. LAI values typically change dynamically between successive crop growth stages [9,10]. By monitoring a dense time series of LAI, it is possible to estimate the total yield of a crop several weeks before the actual harvest [11–13]. This allows the farmer to make proactive selective interventions in the crop to increase yield or plan economic aspects of the operation. Selective application of fertilizer only to areas with estimated lower yields is an example of a practice that can save cost and reduce environmental impacts [14].

The era of global satellite monitoring of vegetation began with a series of Landsat satellites launched in succession from the mid-1970s onward, with one satellite replacing another. Exceptionally, however, the highly successful Landsat-5 mission far exceeded its planned lifetime. In any case, the interpretation of Landsat data has typically involved individual images from a particular satellite. This was due both to the limited temporal resolution of Landsat (16 days) and to the limited computational capabilities at the time [15]. A significant increase in observational capacity and a new ability to work with dense time series of satellite observations have been achieved with the launch of Sentinel-2A (2015) and Sentinel-2B (2017), which are part of the Copernicus Earth Observation System operated as a joint European Space Agency and European Commission project [16]. The pair of essentially identical satellites offer a high temporal resolution, typically of 5 days, as a result of their orbital positioning [17].

In 2021, Landsat 9 was successfully launched. It, like Sentinel-2, is the technological twin of the older and still operational Landsat 8. Once in routine operation, Landsat 9 will effectively increase the temporal resolution of the Landsat system from 16 to 8 days [18]. An appealing way to further increase the temporal resolution of remote sensing data is to harmonize Landsat and Sentinel-2 data with one another, thereby creating a combined dataset of observations from both Landsat and Sentinel-2. This possibility is currently being addressed by NASA's Goddard Space Flight Center. The creation of a harmonized Landsat–Sentinel-2 (HLS) product involves radiometric calibrations between individual sensors and their mutual co-registration to a single spatial resolution, coordinate system, and grid [19]. This allows us to monitor the evolution of surface reflectance on a nearly daily basis [19].

It is possible to obtain LAI values directly from the surface reflectance time series using empirical models (calibration of reflectance values or their derivatives to ground survey data), e.g., [20], or by inversion of radiative transfer models [21]. An advantage of using radiative transfer models consists in their versatility, as once they are representatively parameterized, they do not need ground survey data [22]. A typical representation of this approach is the Scattering by Arbitrary Inclined Leaves (SAIL) family of models [23].

In this paper, the potential for obtaining dense LAI time series for selected agricultural crops by inverting the radiative transfer model known as PROSAIL was investigated. PROSAIL couples the leaf-level PROSPECT and stand-level SAIL models. The inversion of forward model simulations of crop reflectance was performed using harmonized vegetation indices of Landsat and Sentinel-2 satellite observations. Specifically, the issue of developing a generic algorithm for obtaining LAI values using harmonized vegetation indices and machine learning methods was addressed.

## 2. Materials and Methods

### 2.1. Reference In Situ Data Measurements

Eight in situ sampling campaigns were organized during the 2017 (3 campaigns) and 2018 (5 campaigns) growing seasons in the lowlands of the Central Bohemia Region known as Polabí (Elbeland). This is one of the most fertile and productive areas in the Czech Republic, with an average annual temperature of 8–9 °C and average annual precipitation of 500–600 mm. The in situ data were used for (1) parameterization of the PROSAIL radiative transfer model, and (2) validation of crop biophysical variables retrieval. The timing of the campaigns was chosen to cover critical phenophases of the crops. In situ work was carried out on 18 plots (188 reference points) in 2017 and 21 plots (244 reference points) in 2018 at cooperating farms located in the Central Bohemia Region (see Figure 1). The spatial distribution of the sampling transects was defined to capture maximum variability in crop conditions using the Normalized Difference Vegetation Index (NDVI) layer extracted from the nearest cloud-free Sentinel-2 or Landsat scene available prior to the planned campaign. Typically, a transect of 10 points at each plot with 40 m spacing was set up.

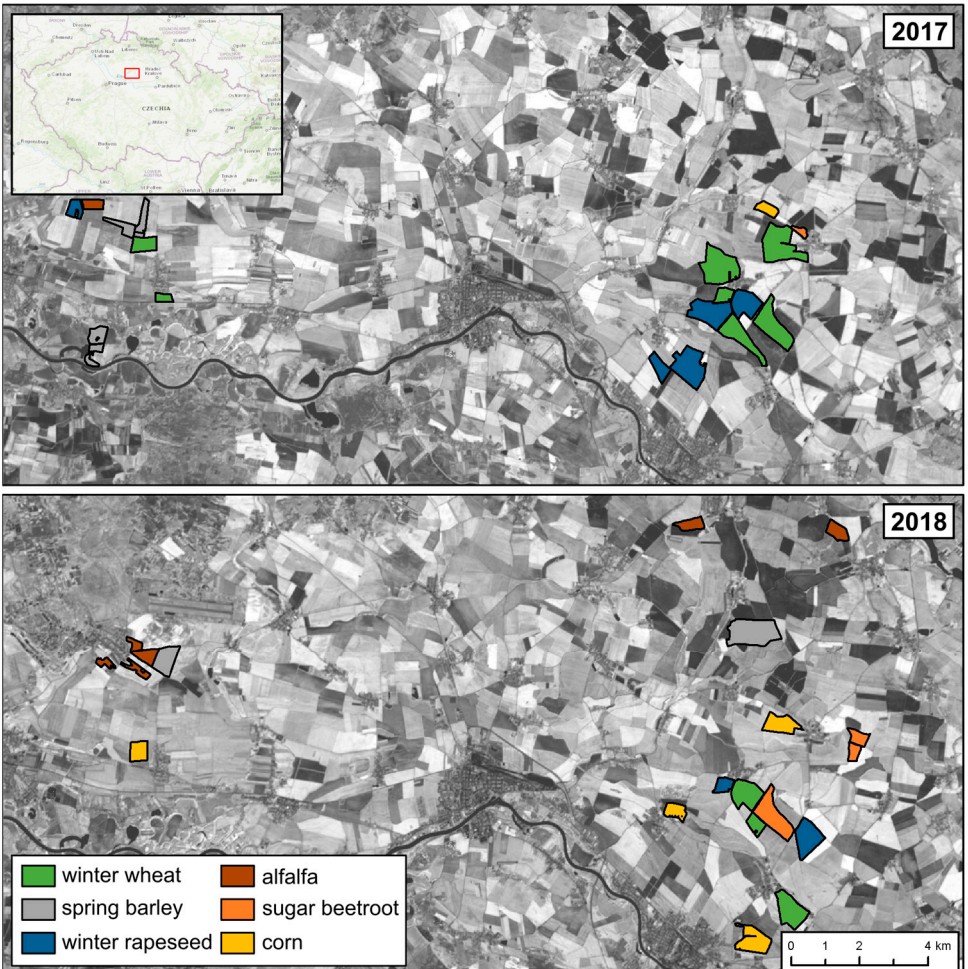

**Figure 1.** Location of agricultural plots where campaigns were conducted in 2017 and 2018, together with the grayscale near-infrared (NIR) band of the Sentinel-2 scene acquired in May of the respective year.

Reference LAI values were obtained (1) using a Delta-T SunScan instrument (Delta-T Devices Ltd.: Cambridge, UK) [24] and (2) from digital hemispherical photography (DHP) using a Canon EOS 700D camera equipped with a Sigma 4.5 mm f/2.8 circular fisheye lens. Five SunScan measurements were taken at each reference point, and their mean was taken as a single reference. SunScan determines LAI by comparing the values of photosynthetically active radiation (400–700 nm) inside the stand with reference values measured outside the stand along its 1 m probe [24]. In addition, eight DHP images were taken at each point using a spatial layout covering an area of $20 \times 20$ m$^2$. The sequences of images were then processed using CanEye software, whereby LAI was retrieved from angular distributions of canopy gaps [25]. Both methods used are suitable for different growth conditions and complement each other: DHP is more suited to low-growing crops (e.g., sugar beet, alfalfa) and for early development stages of crops that can be difficult to measure with SunScan. On the other hand, SunScan works well even in cases of high and dense canopies, where the DHP-based method can underestimate the measurement due to the effect of overlapping leaves in the lower layers of the stand. The range of LAI values obtained during the in situ sampling campaigns is shown in Table 1.

**Table 1.** Summary of LAI values measured in situ.

| Crop | Count | Min | Max | Mean | SD |
|---|---|---|---|---|---|
| Winter wheat | 180 | 0.31 | 6.31 | 3.68 | 1.56 |
| Spring barley | 60 | 0.24 | 7.67 | 4.19 | 1.90 |
| Winter rapeseed | 107 | 0.61 | 8.62 | 3.45 | 2.23 |
| Alfalfa | 57 | 0.09 | 10.16 | 2.78 | 2.48 |
| Sugar beetroot | 62 | 0.86 | 6.72 | 4.33 | 1.66 |
| Corn | 71 | 0.70 | 5.78 | 3.46 | 1.32 |

In addition to LAI, chlorophyll, water, and dry matter contents were also measured at the leaf level. The measured ranges of those other characteristics, along with a detailed description of the measurement methodology, are given in [26]. These measurements were then used to constrain the range of model input parameters and limit the uncertainties in the inversion.

## 2.2. Pre-Processing of Sentinel-2 MSI and Landsat OLI Imagery

The raw (L1) Sentinel-2 and Landsat imagery were used to obtain the LAI of the crops and their changes over time. The images were selected as close as possible to the day of the in situ campaigns with a maximum difference of 5 days (see Table 2). Sentinel-2 images were atmospherically corrected and calibrated to at-surface (top-of-canopy) reflectance (i.e., L2A product) using the SEN2COR processor [27]. Masks of valid pixels corresponding to each scene were then created using the scene classification layer (SCL). All pixels classified as no-data (0), saturated and defective pixels (1), shadows (3), medium probability clouds (8), high probability clouds (9), thin cirrus (10), and snow (11) were considered invalid and excluded from further processing. Similarly, the Landsat data were atmospherically corrected using the Atmospheric and Radiometric Correction of Satellite Imagery (ARCSI) algorithm [28], which calibrates the original pixel values to the surface (top-of-canopy) reflectance. Clouds (and other invalid pixels such as shadows, snow, etc.) were masked using the FMask algorithm [29–31]. The quality of the atmospheric corrections of Sentinel-2 images performed was verified in [26].

**Table 2.** Dates of in situ data collection campaigns and acquisitions of reference Sentinel-2 MSI and Landsat OLI images.

| In Situ Campaign Date | Reference Sentinel-2 Scene Acquisition Date | Reference Landsat Scene Acquisition Date |
|---|---|---|
| 29–31 March 2017 | 1 April 2017 | 1 April 2017 |
| 17–19 May 2017 | 14 May 2017 and 21 May 2017 | 19 May 2017 |
| 19–21 June 2017 | 20 June 2017 | 20 June 2017 |
| 4–5 April 2018 | 6 April 2018 | Not Available |
| 27–30 April 2018 | 26 April 2018 | 28 April 2018 |
| 21 May 2018 | 21 May 2018 | 22 May 2018 |
| 20–21 June 2018 | 20 June 2018 | Not Available |
| 26 July 2018 | 28 July 2018 | 25 July 2018 |

## 2.3. Leaf Area Index Retrieval Approach

The proposed approach of estimating the Leaf Area Index from Sentinel-2 MSI and Landsat OLI data uses the PROSAIL radiative transfer model to generate spectral simulations of crop stands (stored in look-up tables). The PROSAIL spectral simulations are further harmonized by deriving a set of vegetation indices and used as training data in the regression model.

### 2.3.1. Generation of Look-Up Tables

The applied look-up tables generation approach uses crop-optimized PROSAIL models developed in [26]. Focused on quantitative estimation of the LAI, leaf chlorophyll content

(LCC), and leaf water content (LWC) parameters of agricultural crops from Sentinel-2 data [26]. Optimization of the PROSAIL model (reduction of input variability) was performed using the calibration part of the in situ dataset. For each crop of interest, the model was run in forward mode and six look-up tables were created, each containing a total of 550,000 simulations supplemented with a set of biophysical and biochemical parameters used. The PROSAIL input parameters of each simulation were randomly generated with either uniform distribution (for directly measured parameters: LAI, LCC, LWC, specific leaf weight [SLW]), or an empirical distribution (for optimized parameters: N, LIDF_A, LIDF_B, HOTSPOT) within the range of values for the specific crop shown in Table A1. The SKYL parameter (fraction of diffuse solar radiation) depends upon atmospheric and solar illumination conditions but was fixed at 0.2 (the value for cloudless sky [32]) because it has only a limited effect on canopy reflectance [33]. The ranges of solar zenith and azimuth angles were derived from their annual course in the Czech Republic at local solar noon for each day of the year. The variability of soil background spectra was introduced by spectral mixing of dry- and wet-soil spectra, which are part of the PROSAIL model [34]. Mixing was performed at 11 different levels, from completely dry (100% dry soil spectra) to completely wet (100% wet soil spectra) soil. The simulated crop spectra were finally resampled from the original 1 nm spectral resolution to the spectral resolution of the satellite sensor using the sensor response function [35,36].

### 2.3.2. Harmonization of Vegetation Indices across Satellite Systems

A two-step procedure to harmonize the Sentinel-2 MSI and Landsat OLI spectral data that could be used in a single, general LAI prediction model was proposed. The first step was to convert the original reflectances into a set of spectral (vegetation) indices (VIs). A key criterion for the selection of VIs was to retain information on the most important vegetation parameters influencing the spectral properties of the crops: leaf chlorophyll and water content and the total biomass, e.g., [37–39]. The limited number of Landsat spectral bands was also an important factor in the selection of VIs. The absence of red edge wavebands in Landsat, for example, did not allow for using chlorophyll indices such as the Canopy Chlorophyll Content Index (CCCI). The vegetation indices NDVI and Modified Soil Adjusted Vegetation Index (MSAVI) are widely used as proxies for LAI calculations because they have strong relationships with green biomass and complement each other [40,41]. While NDVI performs well for the late developmental stages of crops with full cover, MSAVI has a stronger relationship with biomass in the early phenological stages, where the influence of soil reflectance is pronounced [42]. The frequently used Normalized Difference Water Index with central wavelength 1610 nm (NDWI_1610) was chosen as a suitable VI for determining leaf water content [43]. By selecting such a set of vegetation indices, crop LAI was predicted using two of what can be termed "traits," these being (1) leaf biomass (using proxies NDVI and MSAVI), and (2) foliage water content (using proxy NDWI_1610). The aforementioned VIs were calculated according to the following formulas:

$$\text{NDVI} = (\text{NIR} - \text{RED})/(\text{NIR} + \text{RED}) \qquad (1)$$

$$\text{MSAVI} = (2 \times \text{NIR} + 1 - \text{SQRT}((2 \times \text{NIR} + 1)^2 - 8 \times (\text{NIR} - \text{RED})))/2 \qquad (2)$$

$$\text{NDWI\_1610} = (\text{NIR} - \text{SWIR\_1610})/(\text{NIR} + \text{SWIR\_1610}) \qquad (3)$$

where the RED (visible red band) corresponds to Sentinel-2 MSI band 4 and Landsat OLI band 4, NIR (near-infrared band) corresponds to Sentinel-2 band 8A and Landsat band 5, and SWIR_1610 corresponds to Sentinel-2 band 11 and Landsat band 6. The wavebands upon which the selected VIs are based have comparable spectral characteristics for both types of sensors (Sentinel-2 MSI and Landsat OLI), as can be seen from the response functions in Figure 2.

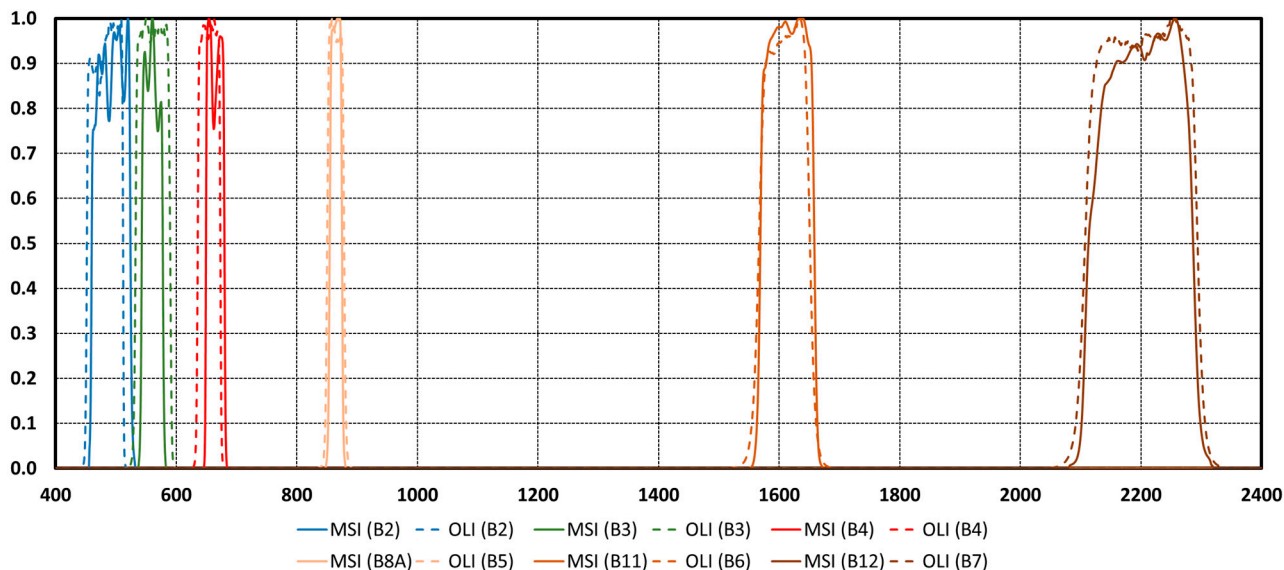

**Figure 2.** Spectral response functions of Sentinel-2 MSI and Landsat OLI common bands.

The second step in harmonization was the application of a simple linear transformation of the LS-VIs (Landsat OLI-derived spectral vegetation indices) to S2-VIs (Sentinel-2 MSI-derived spectral vegetation indices) using a linear regression function. The transformation coefficients were calculated by comparing 129,718 pairs of LS-VI and S2-VI values derived from Sentinel-2 MSI and Landsat OLI top-of-canopy spectral data acquired on the same days during the 2017–2020 growing seasons. The spatial distribution of the sample points was randomly generated to evenly cover all the crops of interest in different climatic regions of the Czech Republic. Visualization of the scatterplots for the MSAVI, NDVI, and NDWI_1610 datasets is shown in Figure 3. Linear transformations were performed using the following formulas:

$$S2\_NDVI = 1.0271 \times LS\_NDVI - 462.68 \tag{4}$$

$$S2\_MSAVI = 0.9884 \times LS\_MSAVI - 149.32 \tag{5}$$

$$S2\_NDWI\_1610 = 0.9958 \times LS\_NDWI\_1610 - 338.49 \tag{6}$$

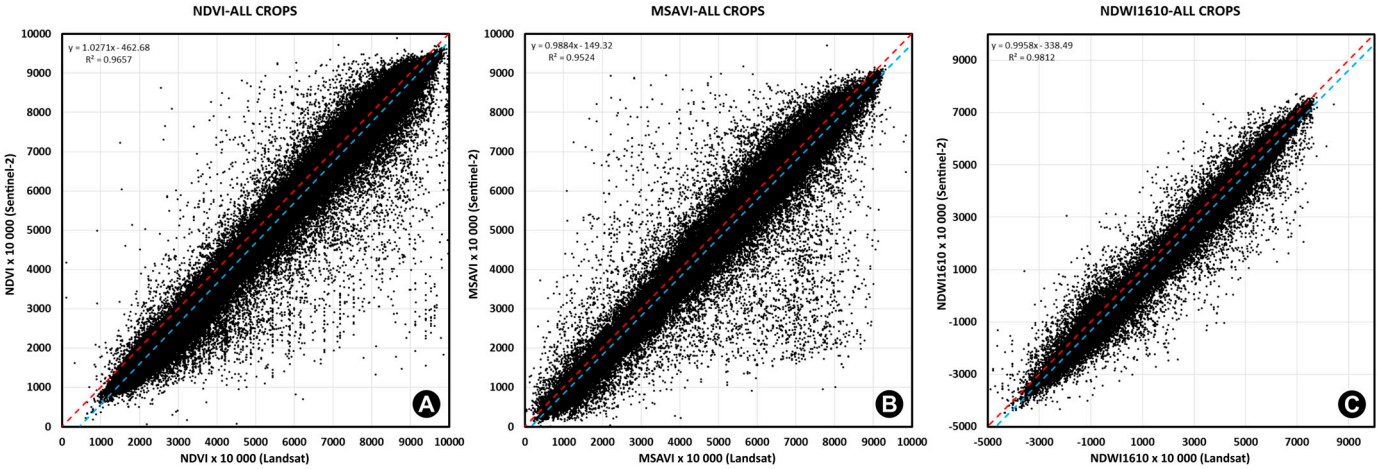

**Figure 3.** Scatter plots of spectral vegetation indices derived from Sentinel-2 MSI and Landsat OLI and calculated for agricultural crops of interest during the period 2017–2020 across the entire Czech Republic: NDVI (**A**); MSAVI (**B**); and NDWI_1610 (**C**).

### 2.3.3. Regression Model

In the present study, the artificial neural network (ANN) approach was used as the regression model to maintain methodological consistency with [26], where the ANN algorithm was also used for LAI estimation and upon which this study builds. The main advantages of the ANN approach include the ability to implicitly model complex nonlinear relationships and thus reveal different interactions between input and output variables [44]. Its main disadvantage is its "black box" nature, inasmuch as the principle of network training is not known [45]. This originates from the very principle of training neurons using the gradient descent method and optimizing the weight function of each neuron [46].

A feed-forward neural network with one hidden layer was designed and implemented using the TensorFlow 1.10.1 machine learning library in Python programming language. The widely used rectified linear unit (ReLu) was chosen as the activation function in the hidden layer neurons, e.g., [47,48]. The training dataset was divided into calibration and validation subsets in a ratio of 8:2. The maximum set number of epochs (100) was never reached due to early stopping mechanisms implemented to prevent overfitting (training was stopped after the mean error loss function reached a baseline value of 0.1 and no improvement was observed in the following five epochs).

### 2.3.4. Validation of Results

An assessment of the accuracy of LAI estimates from Sentinel-2 and Landsat (described in Sections 3.1 and 3.2) was made, based upon the validation part of the in situ dataset. LAI retrieval was always performed for the cloud-free image closest to the field campaign, with a maximum possible delay of 5 days between ground-based data collection and image acquisition. The in situ measurement dates, together with Sentinel-2 and Landsat scenes that were used, are given in Table 2. Accuracy was evaluated using four validation metrics: root mean square error (*RMSE*), its relative expression (*rRMSE*), Pearson's correlation coefficient (*r*), and the coefficient of determination ($R^2$).

LAI estimations from Sentinel-2 MSI and Landsat OLI data were also compared with each other. The comparison was performed on the validation part of the in situ dataset for all campaign data for which a corresponding Sentinel-2 and Landsat image was taken preferably on the same day was available. Again, *RMSE*, *rRMSE*, *r*, and $R^2$ were calculated. The entire harmonized LAI calculation workflow was applied to the time series of Sentinel-2 and Landsat images to demonstrate the temporal and spatial consistency of estimates from both types of sensors.

## 3. Results

### 3.1. Harmonized Sentinel-2 and Landsat-Based Vegetation Indices

Using a general regression model, the harmonized Sentinel-2 MSI (S2-VI) and Landsat OLI (LS-VI) vegetation index datasets were compared to test their substitutability in LAI estimation. Coefficients of determination ($R^2$) reached 0.97, 0.95, and 0.98 for NDVI, MSAVI, and NDWI_1610, respectively. Using the Student's paired *t*-test at significance level $\alpha$ = 0.05, it was verified that the values of the harmonized LS-VI datasets do not differ from the S2-VI datasets (*p*-values: 0.9456, 0.9999, and 0.9856, respectively, for NDVI, MSAVI, and NDWI_1610). Harmonized datasets of vegetation indices are visualized using box plots in Figure 4.

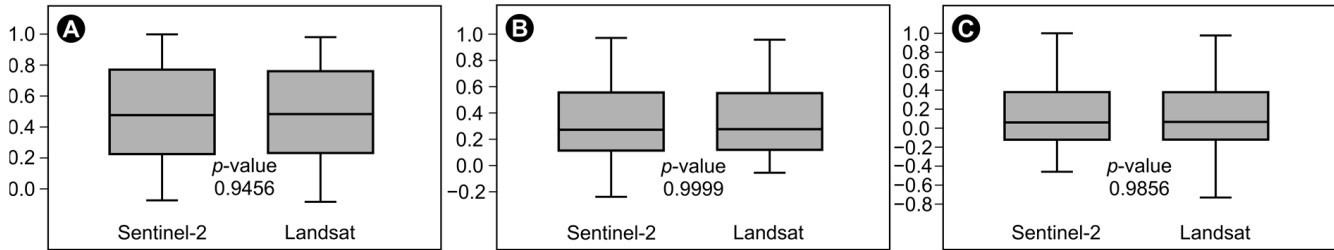

**Figure 4.** Box plots of S2 and LS harmonized VI datasets: NDVI (**A**); MSAVI (**B**); and NDWI (**C**).

### 3.2. Validation of LAI Quantitative Estimation from Sentinel-2 MSI and Landsat OLI Spectra

The best results in terms of the absolute accuracy of LAI estimates (*RMSE* < 1) were obtained for spring barley, sugar beetroot, and corn with Sentinel-2 data, and with Landsat data, the best results were for winter wheat, alfalfa, sugar beetroot, and corn. On the other hand, somewhat poorer accuracy was achieved for winter rapeseed (*RMSE* = 2.34 and 2.38 for Sentinel-2 and Landsat, respectively). In the case of LAI derived from Sentinel-2, a strong relationship between predicted and reference values was documented for all crops of interest: the lowest $R^2$ was 0.70 (alfalfa) and the highest 0.94 (spring barley). In the case of Landsat, a significant relationship between prediction and reference was also documented for all crops of interest. The lowest $R^2$ value, at 0.48 for spring barley, was rather lower than in the case of Sentinel-2. For alfalfa and sugar beet, on the other hand, the resulting $R^2$ values for Landsat were very high (0.95 in both cases). Values of the validation metrics at the crop level are given in Table 3.

**Table 3.** Validation metrics of Landsat and Sentinel-2 LAI estimations against in situ data on the level of individual crops.

| Crop | S2-LAI | | | | LS-LAI | | | |
|---|---|---|---|---|---|---|---|---|
| | *RMSE* | *rRMSE* | *r* | $R^2$ | *RMSE* | *rRMSE* | *r* | $R^2$ |
| Winter wheat | 1.28 | 0.40 | 0.91 | 0.82 | 0.96 | 0.26 | 0.82 | 0.67 |
| Spring barley | 0.92 | 0.28 | 0.97 | 0.94 | 1.36 | 0.32 | 0.69 | 0.48 |
| Winter rapeseed | 2.34 | 0.75 | 0.89 | 0.79 | 2.38 | 0.77 | 0.85 | 0.73 |
| Alfalfa | 1.43 | 0.50 | 0.84 | 0.70 | 0.83 | 0.20 | 0.98 | 0.95 |
| Sugar beetroot | 0.80 | 0.18 | 0.90 | 0.80 | 0.55 | 0.16 | 0.98 | 0.95 |
| Corn | 0.70 | 0.21 | 0.87 | 0.76 | 0.82 | 0.21 | 0.87 | 0.75 |

### 3.3. Harmonized Sentinel-2 and Landsat-Derived LAI Series

In comparing LAI estimations from Sentinel-2 MSI and Landsat OLI data, the highest absolute error was calculated for alfalfa (*RMSE* = 0.86), which in relative terms represented just 21% (LAI from Sentinel-2 was considered as the reference dataset in this case). For all crops of interest, a significant relationship between the different LAI datasets was also documented ($R^2$ was even higher than 0.9 for winter and spring barley, winter rapeseed, alfalfa, and sugar beetroot). The $R^2$ for corn was 0.56 (Table 4).

**Table 4.** Comparison of Landsat and Sentinel LAI estimations using *RMSE*, *rRMSE*, *r*, and $R^2$ statistics.

| Crop | *RMSE* | *rRMSE* | *r* | $R^2$ |
|---|---|---|---|---|
| Winter wheat | 0.63 | 0.14 | 0.98 | 0.96 |
| Spring barley | 0.59 | 0.14 | 0.95 | 0.90 |
| Winter rapeseed | 0.56 | 0.11 | 0.97 | 0.94 |
| Alfalfa | 0.86 | 0.21 | 0.96 | 0.93 |
| Sugar beetroot | 0.24 | 0.07 | 0.99 | 0.98 |
| Corn | 0.49 | 0.15 | 0.75 | 0.56 |

Seasonal trajectories visualized in Figure 5 were created by the spatial aggregation of harmonized LAI time series covering in situ plots (beginning of the year to the end of October 2018) at the level of individual crops. Sentinel-2 and Landsat acquisitions are color coded. The graphs show that the temporal consistency of the harmonized time series is good in all cases.

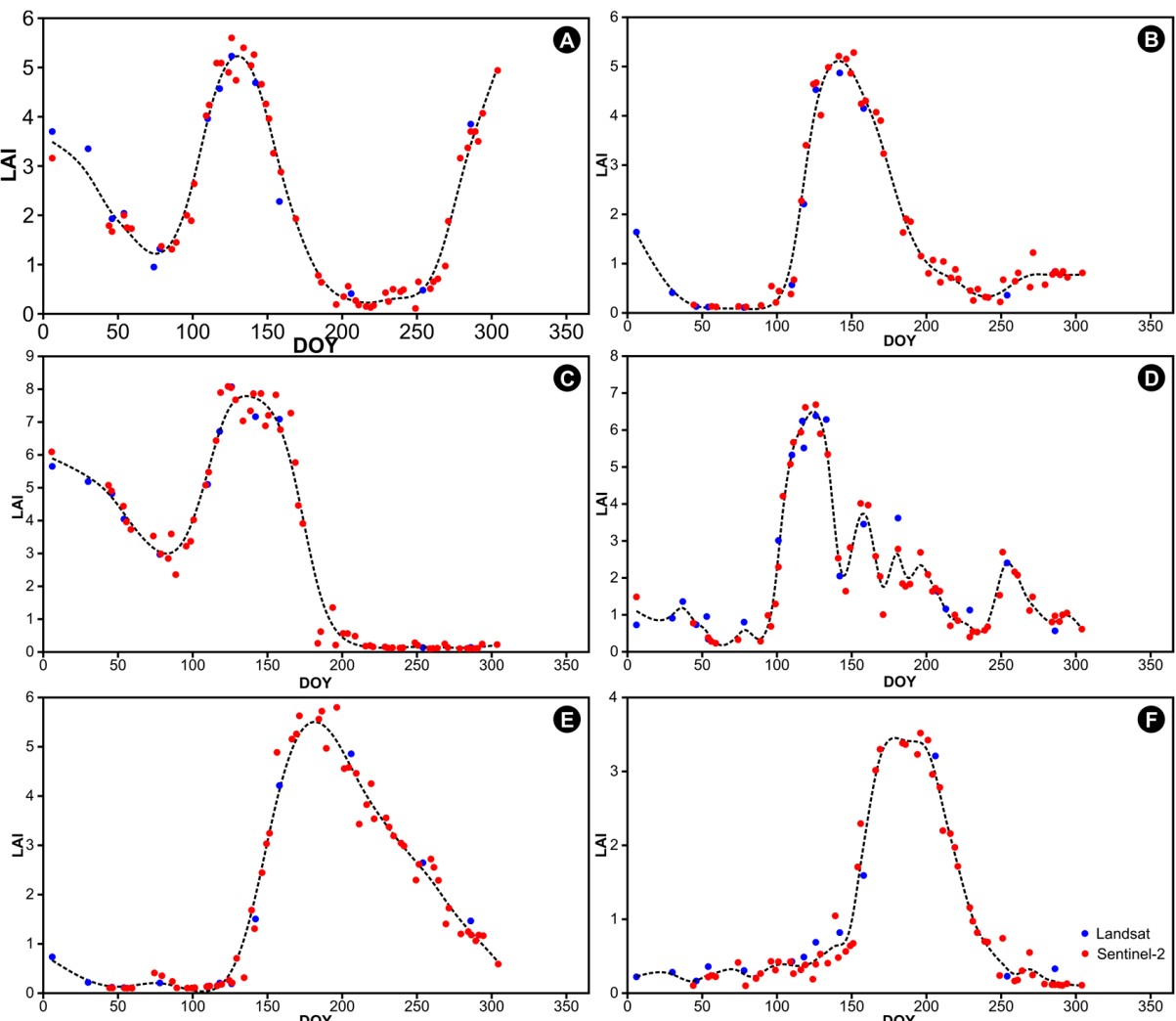

**Figure 5.** Example of harmonized LAI time series from Sentinel-2 MSI and Landsat OLI data acquired during 2018 for winter wheat (**A**); spring barley (**B**); winter rapeseed (**C**); alfalfa (**D**); sugar beetroot (**E**); and corn (**F**).

To demonstrate the spatial consistency of the LAI layers derived from Sentinel-2 and Landsat data, three image pairs from 2017 (Sentinel-2 and Landsat 8) were selected. In two cases, the images from both sensors were acquired on the same day (1 April and 20 June 2017). In one case, the delay between acquisitions from each sensor was 2 days (14 May 2017 for Sentinel-2 and 16 May 2017 for Landsat 8. The derived LAI layers are shown in Figure 6. Across all the dates shown, a strong agreement in the spatial patterns of LAI, either at the individual plot level or when comparing several plots with different vegetation conditions, can be observed.

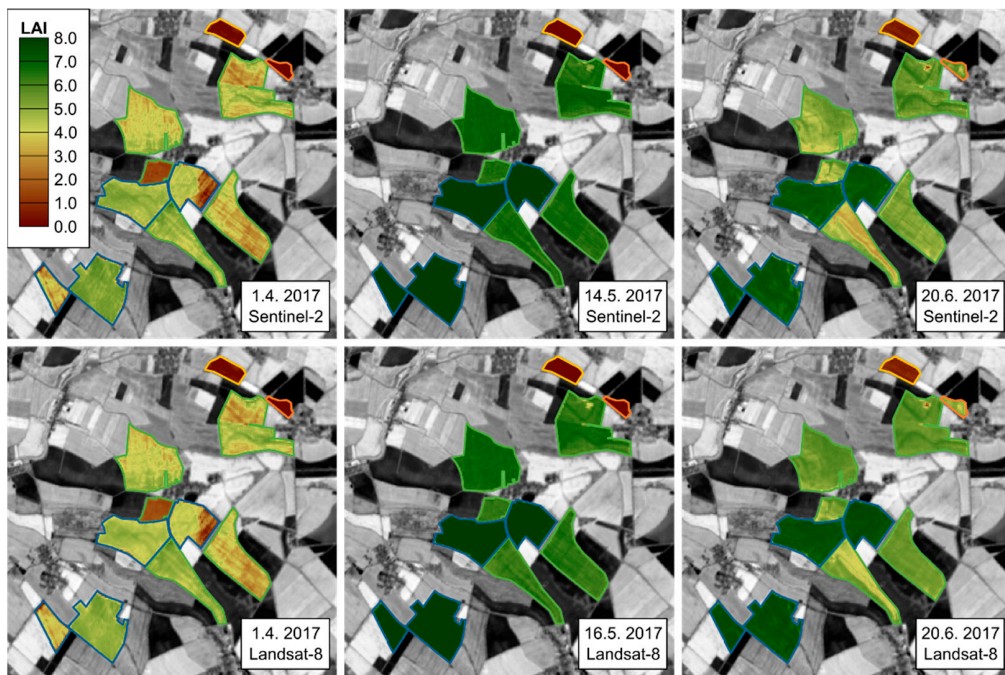

**Figure 6.** Examples of Landsat 8 and Sentinel-2 LAI layer pairs derived from the near acquisition date images. Plot borders are drawn in color according to crop type. Winter wheat (green), winter rapeseed (blue), sugar beetroot (orange), and corn (yellow) grow on the plots shown.

## 4. Discussion

In this paper, the successful acquisition of LAI values for a dense time series of observations from two satellite sensors, Landsat OLI and Sentinel-2 MSI, was demonstrated. In general, a very strong agreement was observed between the LAI obtained and its independent ground-based measurement ($R^2$ between 0.7 and 0.94 for the Sentinel-2 dataset and between 0.48 and 0.95 for the Landsat dataset Table 4). In the case of Sentinel-2, the strongest agreement was obtained for cereals and sugar beet ($R^2 > 0.8$) and the weakest, though still very conclusive, for alfalfa ($R^2 = 0.7$). This may be due to the stand structure itself and the timing of its growth: while cereals typically form dense, closed stands with peak foliage in late spring, alfalfa is smaller in stature and sometimes is used as an intercrop. Alfalfa's canopy cover may thus be smaller and the effect of, for example, soil compaction greater. Turning to *RMSE*, the obtained results for alfalfa are comparable to those achieved by [21] using pixel-based inversion of the PROSAIL model. Winter wheat LAI retrieval from Sentinel-2 imagery based on the PROSAIL model and ANN or look-up table (LUT) was applied by [49]. Their validation with in situ LAI measurements revealed $R^2$ of 0.5 and *RMSE* of 1.35 and 1.60 for the ANN and LUT methods, respectively. As indicated by these values, the developed model performed better for both types of input imagery.

In the present study, slightly different results were observed when comparing the Sentinel-2 and Landsat-based LAI validation metrics. In the case of Landsat and in contrast to Sentinel-2, the best agreement was achieved for sugar beetroot and alfalfa ($R^2 = 0.95$) and the poorest ($R^2 = 0.48$) for spring barley. These differences might be attributed to different temporal coverage of the two instruments causing different temporal patterns with respect to the phenology of the crops as well as slight time shifts between in situ measurements and satellite data acquisition. Ref. [50] had studied differences in LAI and biomass retrieval from the Harmonized Landsat Sentinel-2 Surface Reflectance product (HLS) of six crops (canola, soybean, wheat, corn, oat, and beans) in southern Manitoba, Canada. Using linear and nonlinear regression models relating vegetation indices and LAI for all the crops pooled together, their comparison with in situ data revealed the best results for the Modified Simple Ratio Red-edge index [51], with *RMSE* of 0.38 and *rRMSE* of 0.18 and the Normalized Difference Water Index [52], with *RMSE* of 0.83 and *rRMSE* of 0.20 for the

Sentinel-2 and Landsat imagery, respectively. Though the present experiment performed slightly less well, it can be seen as a next step toward a more generalized approach using the same workflow and metrics for LAI retrieval from both types of data.

The retrieval of LAI values for the Landsat OLI and Sentinel-2 MSI sensors are comparable to one another and so, too, are the ranges of values and measurement errors for the two systems (Table 4, Figures 5 and 6). This is somewhat to be expected inasmuch as the applied approach to the retrieval of LAI values uses harmonized vegetation indices utilizing comparable wavelengths for the two satellite systems (see Figure 2). In this way, it was possible to reduce differences between the LAI values obtained from the two sensors, as mentioned in [50], especially around the peak of the season. On the other hand, a lower spatial resolution of 30 m for the Landsat OLI satellite system was combined with the 20 m resolution of the Sentinel-2 satellite. Thus, part of the difference in the retrieval of LAI values may be due to the spectral mixture at the 20 m vs. 30 m pixel. Other differences may be due to the harmonization of the two datasets. In the present study, a simple approach to harmonizing vegetation indices was used. In principle, a linear function developed over a representative dataset of observations through the growing season was applied. A possible improvement might be achieved by spectral harmonization of the source digital number (DN) values or reflectance values and spatial harmonization of the datasets to the same resolution, coordinate system, and point spread function. This is addressed, for example, by the Harmonized Landsat Sentinel product [19]. A further study [50] also reported the greatest accuracy of biomass estimation when based upon the HLS product.

The applied approach harmonizing vegetation indices produces a high degree of agreement in the LAI values obtained (typical $R^2$ between 0.9 and 0.98), an exception being for corn, which had a lower $R^2$ value of 0.56, albeit with *RMSE* and *rRMSE* values similar to those for other crops. Although this indicates the good performance of this approach, it should be noted that when it is applied to other crops or other vegetation indices, it always will be necessary to empirically recalibrate the harmonization functions (see Section 2.3.2). The advantages of using the harmonized vegetation indices of the Sentinel-2 and Landsat satellite systems are that it increases the probability of obtaining an LAI value and can densify the time series of observations. This is particularly important for the early stages of crop development, during which high cloud cover and reduced availability of observations from optical satellite systems are typically encountered. With the availability of Landsat 9 data, these possible time series of harmonized vegetation indices will be further condensed. That can contribute to the earlier estimation of future yield and early detection of sites that will benefit from selective fertilizer application.

## 5. Conclusions

Sentinel-2 MSI and Landsat OLI multispectral images provide a valuable source of data for monitoring the condition of agronomic crop stands by deriving various vegetation indices or biophysical parameters. One of the most interesting biophysical parameters in terms of its use in so-called precision agriculture is the Leaf Area Index, describing the amount of biomass within a stand, which, in turn, directly influences the total resulting yield. The efficiency of agricultural monitoring using remote sensing data increases rapidly with a rising density of observations, as it is possible to capture even gradual changes in the vegetation during key phases of the growing season. The technological twins of the Sentinel-2 (A/B) and Landsat (8/9) satellites provide similar data (in terms of spectral and spatial resolution) with a relatively high temporal resolution of data acquisition (ca. 5 days for Sentinel-2 and ca. 8 days for Landsat). By harmonizing products derived from Sentinel-2 and Landsat data, their potential for agricultural monitoring can be significantly enhanced.

This study addresses the harmonized calculation of LAI from Sentinel-2 MSI and Landsat OLI data. The methodology was designed and tested for the six agronomic crops most frequently cultivated in the Czech Republic: winter wheat, spring barley, winter rapeseed, alfalfa, sugar beetroot, and corn. A general regression model was designed using an artificial neural network and training data was generated using the PROSAIL

radiative transfer model. The harmonization of Sentinel-2 MSI and Landsat OLI input data was based upon the calculation of the vegetation indices NDVI, MSAVI, and NDWI_1610, representing amounts of green biomass and foliage water content. These are vegetation parameters most influencing the spectral response of the stand.

LAI estimates from both types of sensors were validated against in situ data. In general, a very good agreement between the LAI gain and the corresponding ground measurement ($r > 0.7$) was observed. The intercomparison of LAI derived from Sentinel-2 and Landsat data also showed a very strong agreement ($r$ ranging between 0.7 and 0.98) and a low relative *RMSE* value (<20%). Using examples of harmonized LAI time series, high consistency of estimates from both Sentinel-2 and Landsat data over the entire growing season was demonstrated. The agreement of the spatial patterns was then illustrated by comparing the LAI maps derived from very close acquisitions of both types of satellites.

The main contribution of this study is that it outlines a generic yet crop-specific model for deriving a harmonized LAI time series combining Sentinel-2 and Landsat imagery as input data. Such a condensed time series of observations will be highly useful in effective agricultural management, including, for example, for accurately determining the timing of selective fertilization, in harvest planning, and in the early estimation of future yields.

**Author Contributions:** Conceptualization, J.T. and J.M.; methodology, J.T., J.M. and P.L.; software, J.T.; validation, J.T. and J.M.; formal analysis, J.T.; investigation, J.T. and J.M.; resources, J.T., P.L. and M.P.; data curation, J.T.; writing—original draft preparation, J.T., J.M., P.L. and M.P.; writing—review and editing, P.L. and M.P.; visualization, J.T. and J.M.; supervision, J.T. and P.L.; project administration, J.T.; funding acquisition, J.T. All authors have read and agreed to the published version of the manuscript.

**Funding:** This research was funded by the Technology Agency of the Czech Republic, grant number TH02030248: "Use of Copernicus satellite data for an effective monitoring of a status and management of selected agricultural crops" and the Charles University Grant Agency, grant number 293121: "Long-term development of phenological parameters of agricultural crops based on a 35-year time series of Landsat multispectral data". The APC was funded by the Charles University Grant Agency, grant number 293121.

**Institutional Review Board Statement:** Not applicable.

**Data Availability Statement:** Not applicable.

**Acknowledgments:** We would like to thank the agronomists of the cooperating farms for their support in selecting and allowing access to agricultural parcels in order to allow in situ data collection, namely Martina Martincová (Poděbradská Blata a.s.) and Jiří Hlaváček (Pěstitel Stratov s.r.o.). We would also like to thank all those who contributed to the realization of field campaigns, namely Jan Foldyna, Karel Klem (GCRI), Lukáš Fajmon (GCRI), Tomáš Purket (GCRI), Rahul Raj (GCRI), Lucie Brabcová (Gisat Ltd.), Kateřina Tučková (Gisat Ltd.) and Markéta McEwan (Gisat Ltd.). We are also grateful for the financial support of the project to the Charles University Grant Agency (grant number 293121).

**Conflicts of Interest:** The authors declare no conflict of interest. The funders had no role in the design of the study; in the collection, analyses, or interpretation of data; in the writing of the manuscript; or in the decision to publish the results.

## Appendix A

**Table A1.** Ranges of the PROSAIL model inputs used for the generation of look-up tables (LCC = leaf chlorophyll content, LWC = leaf water content; LAI = leaf area index; SLW = specific leaf weight; SA, SZ = solar azimuth and zenith angles; SKYL = fraction of diffuse solar radiation; SWR = soil wetness ratio. N, LIDFa, LIDFb, and HSpot are leaf and canopy structural parameters.).

| Crop | | LCC | LWC | LAI | SLW | N | LIDFA | LIDFB | HSPOT | SA | SZ | SKYL | SWR |
|---|---|---|---|---|---|---|---|---|---|---|---|---|---|
| | | $\mu g/cm^2$ | cm | $m^2/m^{-2}$ | $g/cm^2$ | - | - | - | m/m | deg. | deg. | - | - |
| Winter wheat | **min** | 0 | 0.0005 | 0 | 0.0009 | 1.44 | −1 | −1 | 0.01 | 150 | 25 | 0.2 | 0 |
| | **max** | 80 | 0.07 | 8 | 0.0197 | 1.44 | 0 | 1 | 0.5 | 170 | 70 | 0.2 | 1 |
| Spring barley | **min** | 0 | 0.0005 | 0 | 0.001 | 1.57 | −1 | −1 | 0.01 | 150 | 25 | 0.2 | 0 |
| | **max** | 80 | 0.07 | 8 | 0.0138 | 1.57 | 0 | 1 | 0.5 | 170 | 70 | 0.2 | 1 |
| Winter rapeseed | **min** | 0 | 0.0005 | 0 | 0.0005 | 1.78 | −1 | −1 | 0.5 | 150 | 25 | 0.2 | 0 |
| | **max** | 80 | 0.07 | 10 | 0.01 | 1.78 | 1 | 1 | 0.5 | 170 | 70 | 0.2 | 1 |
| Alfalfa | **min** | 0 | 0.0005 | 0 | 0.003 | 1.53 | −1 | −1 | 0.01 | 150 | 25 | 0.2 | 0 |
| | **max** | 80 | 0.07 | 10 | 0.008 | 1.53 | 1 | 1 | 0.5 | 170 | 70 | 0.2 | 1 |
| Sugar beetroot | **min** | 0 | 0.0005 | 0 | 0.003 | 1.67 | −1 | −1 | 0.1 | 150 | 25 | 0.2 | 0 |
| | **max** | 80 | 0.07 | 8 | 0.008 | 1.67 | 0 | 1 | 0.5 | 170 | 70 | 0.2 | 1 |
| Corn | **min** | 0 | 0.0005 | 0 | 0.003 | 1.28 | −1 | −1 | 0.2 | 150 | 25 | 0.2 | 0 |
| | **max** | 80 | 0.07 | 8 | 0.008 | 1.28 | 1 | 1 | 0.5 | 170 | 70 | 0.2 | 1 |

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
