# Peer review of "Retrieval of Harmonized LAI Product of Agricultural Crops from Landsat OLI and Sentinel-2 MSI Time Series"

_agriculture, doi:10.3390/agriculture12122080_

Round 1
Reviewer 1 Report
This article’ Retrieval of harmonized LAI product of agricultural crops from 2 Landsat OLI and Sentinel-2 MSI time series’ used Sentinel-2 MSI and Landsat OLI multispectral satellite data to propose an approach for harmonized calculation of LAI for agronomic crops dominant in the Czech Republic. The purpose of the research seems to be meaningful. However, the description of the method section is still unclear and needs further improvement. I would recommend to reconsider it for publication after another round of review.
Please see the attachment for details.

Author Response
Response to Reviewer 1 Comments
General comments:
This article’ Retrieval of harmonized LAI product of agricultural crops from 2 Landsat OLI and Sentinel-2 MSI time series’ used Sentinel-2 MSI and Landsat OLI multispectral satellite data to propose an approach for harmonized calculation of LAI for agronomic crops dominant in the Czech Republic. The purpose of the research seems to be meaningful. However, the description of the method section is still unclear and needs further improvement. I would recommend to reconsider it for publication after another round of review.
First, the article lacks a more detailed description of the PROSAIL model. Also, it is recommended to add a flow chart in the Method section to help the readers understand the entire method flow. Furthermore, I cannot clearly understand how the look up table generated by the PROSAIL model was used in the method. In addition, how did the generated look up table interact with the ANN algorithm to invert the LAI of remote sensing imagery? Did the authors developed the regression model using in-situ measured LAI and the corresponding remote sensing-based vegetation indices as inputs? The authors should explain the method more thoroughly and described the logical relationship between the parts of the method.
Response: Thanks for your feedback. We have added a brief paragraph to section 2.3 Leaf Area Index Retrieval Approach describing the interconnection of the PROSAIL model, look-up tables and the ANN algorithm. Hopefully the methodological approach will now be clearer for the readers.
Specific comments:
P6 Line 210: Why did the authors want to maintain methodological consistency with their previous work? The authors should elaborate on the reasons why the method must be consistent.
Response: Thanks for your point. The main reason was the use of experience from previous research in which a LAI estimation model (among other parameters) from Sentinel-2 data was proposed. The argumentation was also modified in the manuscript (section 2.3.3.).
Figure 4: It is recommended to change Figure 4 as a scatter plot. Also, the authors should confirm that the obtained p-values are correct. Because in general, the assumption of t-test is that the two variables are not correlated. Therefore, the smaller the p-value, the stronger the correlation.
Response: Thanks for your feedback. We believe that the use of a boxplot is preferable in this case for two reasons: first, the scatter plots for individual vegetation indices "before harmonization" are already shown in Figure 3, and the visual change of the harmonized scatter plots will be greatly limited. Secondly, with Figure 4 we document that the individual datasets of vegetation indices are not significantly different (not a comparison of the mutual relationship between the two datasets). Similarly, with the paired t-test: we do not evaluate the correlation, but the statistical significance of the differences between the mean values of two paired datasets.
Reviewer 2 Report
Hi Authors
Please find attached the document with comments
Regards
Anonymous

Author Response
Response to Reviewer 2 Comments
Thank you for your constructive recommendations indicated in the form of comments directly in the manuscript. We've incorporated the vast majority of these suggestions (and highlighted them with "track changes"). For the recommendation to move parts of the results to the Materials and methods chapter, we would like to provide a brief comment: We partially shortened the paragraphs 3.2 and 3.3 and added the section "2.3.4. Validation of results" in chapter 2 Materials and methods. The text is now clearer and smoother to read, but some essential information had to be preserved also in the chapter Results.
Round 2
Reviewer 1 Report
The authors have presented revisions to justify the Methods section. However, the method is still not clear enough and needs a few minor revisions. The details are in the attachment.

Author Response
Response to Reviewer 1 Comments
First, the description of the establishment of the ANN regression model is still not clear. The authors replied that the crop spectra simulated by the PROSAIL model were used to train the regression model after being harmonized. However, including only spectral or spectrally derived indices, but not LAI, how does the regression model for LAI be built? Did the authors use the harmonized simulated spectra and the LAI values corresponding to each spectrum in the lookup table as input features to build the regression model? If so, please add the corresponding description.
Our comment: Thank you for your point, we have added clarification on the issue of generating training data for the ANN regression model on L160-161.
Second, where is Figure 3? The authors should confirm that Figure 3 is correctly included in the manuscript.
Our comment: Figure 3 should be placed in the manuscript between lines 215 and 216 and was also attached (along with the other figures) as a separate file.